# Biofilm Inhibition and Eradication Properties of Medicinal Plant Essential Oils against Methicillin-Resistant *Staphylococcus aureus* Clinical Isolates

**DOI:** 10.3390/ph13110369

**Published:** 2020-11-06

**Authors:** Fethi Ben Abdallah, Rihab Lagha, Ahmed Gaber

**Affiliations:** 1Department of Biology, College of Sciences, Taif University, P.O. Box 11099, Taif 21944, Saudi Arabia; rihab.k@tu.edu.sa (R.L.); a.gaber@tu.edu.sa (A.G.); 2Unité de Recherche, Virologie & Stratégies Antivirales, UR17ES30, Institut Supérieur de Biotechnologie, University of Monastir, Monastir 5000, Tunisia; 3Department of Genetics, Faculty of Agriculture, Cairo University, Giza 12613, Egypt

**Keywords:** methicillin-resistant *Staphylococcus aureus*, essential oils, *Origanum majorana*, *Rosmarinus officinalis*, *Thymus zygis*, antibacterial, biofilm inhibition and eradication

## Abstract

Methicillin-resistant *Staphylococcus aureus* is a major human pathogen that poses a high risk to patients due to the development of biofilm. Biofilms, are complex biological systems difficult to treat by conventional antibiotic therapy, which contributes to >80% of humans infections. In this report, we examined the antibacterial activity of *Origanum majorana*, *Rosmarinus officinalis*, and *Thymus zygis* medicinal plant essential oils against MRSA clinical isolates using disc diffusion and MIC methods. Moreover, biofilm inhibition and eradication activities of oils were evaluated by crystal violet. Gas chromatography–mass spectrometry analysis revealed variations between oils in terms of component numbers in addition to their percentages. Antibacterial activity testing showed a strong effect of these oils against MRSA isolates, and *T. zygis* had the highest activity succeeded by *O. majorana* and *R. officinalis*. Investigated oils demonstrated high biofilm inhibition and eradication actions, with the percentage of inhibition ranging from 10.20 to 95.91%, and the percentage of eradication ranging from 12.65 to 98.01%. *O. majorana* oil had the highest biofilm inhibition and eradication activities. Accordingly, oils revealed powerful antibacterial and antibiofilm activities against MRSA isolates and could be a good alternative for antibiotics substitution.

## 1. Introduction

Methicillin-resistant *Staphylococcus aureus* (MRSA) is considered a principal human pathogen and the most common cause of nosocomial infections. MRSA causes several diseases ranging from skin and soft tissue infections to serious invasive infections such as pneumonia, bacteremia, endocarditis and osteomyelitis [1]. The number of MRSA infections, which are more frequently associated with mortality than other bacterial infections, has increased considerably over recent years. *S. aureus* carries 20–40% mortality at 30 days despite appropriate treatment [2].

MRSA poses a high risk to patients due to the development of biofilm [3]. Biofilm is considered as major virulence factor and is an organized structure built by almost all bacteria that is composed of nucleic acids, lipids, proteins, and polysaccharides [4]. Biofilms contribute to >80% of human infections and *S. aureus* is considered as the leading species in biofilm-associated infections [5]. In Biofilm, MRSA like other bacteria, become more persistent in the host organism, environment, and medical surfaces, and show an increased resistance to antibiotics and host immune factors [6,7,8], which is an important medical problem. Therefore, the development of novel compounds to treat biofilm is urgently required; plant essentials oils (EOs) that act against bacterial biofilm are of great interest.

EOs are volatile compounds that have been used to combat a variety of infections during hundreds of years as a natural medicine. It has been shown that EOs possess several significant antimicrobial activities such as antibacterial, antiviral, antifungal, and anti-parasitic activities in addition to their antioxidant, antiseptic, and insecticidal properties [9,10].

*Rosmarinus officinalis* L., *Thymus zygis* L., and *Origanum majorana* L. belong to the Lamiaceae family. EOs obtained from aerial parts of the flowering stage of these plants, have been reported for their antibacterial activities against *S. aureus* [11,12] and their antibiofilm activities against uropathogenic *E. coli* [13]. Several reports have shown that tea tree, thyme, and peppermint EOs, are effective against planktonic [14] and biofilm [15,16] MRSA. In addition, Cáceres et al. [17] demonstrated high anti-biofilm activity of thymol-carvacrol-chemotype (II) oil from *Lippia origanoides* against *E. coli* and *S. epidermidis*. However, these oils did not alter the growth rate of planktonic bacteria. The antibacterial effect of EOs, which is manifested by alterations of the bacterial cell wall and cell membrane, depends of their chemical composition [18]. The cell membrane compositions play an important role in the high resistance of Gram-negative bacteria to EOs compared to Gram-positive [19]. The hydrophobic molecules penetrate easily into the cells due to cell wall structure in Gram-positive bacteria and act on the cell wall and within the cytoplasm [20].

This study aimed to investigate the antibacterial, biofilm inhibition, and eradication properties of *O. majorana*, *R. officinalis*, and *T. zygis* medicinal plants’ EOs against clinical methicillin-resistant *Staphylococcus aureus*.

## 2. Results

### 2.1. Distribution of the MRSA Isolates

Thirty clinical MRSA isolates were collected from King Abdulaziz Specialist Hospital, Taif, Saudi Arabia. The isolates were obtained from infection sites: surgical site infection (SSI, *n* = 4), skin and soft tissue (SST, *n* = 12), blood (*n* = 1), nasal (*n* = 8) and burn (*n* = 5). The distribution of isolates based on the type of specimen is presented in Figure 1.

### 2.2. Chemical Composition of the Essential Oils

*O. majorana*, *T. zygis*, and *R. officinalis’* EOs chemical compositions are summarized in Table 1. In total 37 components were detected in these oils: 10 compounds in *R. officinalis* and 31 compounds in each of *O. majorana* and *T. zygis*.

GC-MS results showed variations between these oils regarding the compound numbers and their percentages. The major constituents of *O. majorana* were terpinen-4-ol (25.9%), γ-terpinene (16.9%), linalool (10.9%), sabinene (8%), and α-terpinene (7.7%); those of *R. officinalis* were α-pinene (37.7%), bornyl acetate (9.1%), camphene (7.3%), borneol (5.5%), verbenone (5.4%), camphor (5.2%), and 1,8-cineole (4.7%). However, the main components of *T. zygis* were linalool (39.7%), terpinen-4-ol (11.7%), β-myrcene (8.6%), and γ-terpinene (7.6%).

### 2.3. Antibacterial Activity of Essential Oils against MRSA

#### 2.3.1. Disc Diffusion

The antibacterial activity of EOs against MRSA isolates was assessed by the disc diffusion method (Table 2). *T. zygis* EO has shown strong inhibitory activity on 80% of the strains, succeeded by *O. majorana* and *R. officinalis* that demonstrated a strong inhibitory action on 53.33% and 16.66% of the isolates, respectively. Moreover, according to the high percentage of anti-MRSA activity, *T. zygis* and *O. majorana’* EOs have a strong inhibitory action on 80% and 53.33% of the strains, respectively. However, *R. officinalis* had a slight inhibitory action on 46.66% of the strains. Thereby, *T. zygis’* EO appeared as the EO with the highest antibacterial activity, succeeded by *O. majorana* and *R. officinalis*. According to the type of specimen, globally, the same result was found as detected in cases of total isolates. *T. zygis* was regarded as an EO with strong inhibitory activity, succeeded by *O. majorana* and *R. officinalis*.

#### 2.3.2. Determination of MIC and MBC

Antibacterial activity of EOs was assessed by measuring MICs and MBCs for the 30 MRSA isolates and the reference strain. The values of MIC extended from 0.78 mg/mL to 1.56 mg/mL, while the MBC varied from 3.125 mg/mL to 12.5 mg/mL for *O. majorana* EO.

Concerning *T. zygis* EO, the MIC values ranged from 0.39 mg/mL to 0.78 mg/mL, while the MBC value was 3.125 mg/mL. However, The MIC values of *R. officinalis* varied from 0.78 mg/mL to 1.56 mg/mL, but the MBC value was 12.5 mg/mL. Compared to *O. majorana* and *R. officinalis*, *T. zygis* EO showed the greatest antibacterial activity against MRSA isolates. Person correlation (r) showed no significant correlation between the type of specimen and MICs of oils (*p* > 0.05).

### 2.4. Biofilm Formation

MRSA strains were tested for their potentialities to form biofilm on a polystyrene surface. Table 3 indicated that 96.66% of the isolates were able to form biofilm and were distributed as follow: 40% were highly positive biofilm producers with OD570 values ranged from 1.175 to 3.635, and 56.66% were low-grade positive with OD570 values extended from 0.113 to 0.87. However, out of the 30 isolates only one strain was isolated from the nasal sample was biofilm negative. Reference *S. aureus* ATCC 25923 was considered as a highly positive biofilm producer. Analysis of variance (ANOVA) indicated that there is no significant effect of the specimen on biofilm formation (*p* > 0.05).

### 2.5. Biofilm Inhibition Activity of Essentials Oils

Biofilm inhibitory activities of *O. majorana*, *T. zygis*, and *R. officinalis’* EOs are summarized in Table 4. MRSA isolates that showed a biofilm formation potential were selected for this investigation. In all, 29 isolates considered as highly positive biofilm and low-grade positive biofilm in addition to the reference strain were used.

*O. majorana* EO demonstrated an antibiofilm activity on 89.66% of the isolates (26 strains) in addition to the reference strain. Among them, 11 isolates (39.93%) were passed from low-grade positive to biofilm negative, three highly positive biofilm isolates (10.34%) become low-grade positive, and one strain isolated from nasal samples was changed from highly positive to biofilm negative after treatment with *O. majorana* EO. The percentage of inhibition ranged from 10.29 to 95.91%.

Concerning *R. officinalis* EO, we detected an antibiofilm effect on 79.31% of the isolates (23 strains). Furthermore, two groups of five isolates (17.24%) were changed. The first group was varied from low-grade positive to biofilm negative, and the second group was passed from highly positive to low-grade positive. In addition, the same isolate that changed from highly positive to biofilm negative under *O. majorana* EO, also became biofilm negative under *R. officinalis* EO. The percentage of inhibition extended from 10.20 to 95.65%.

Antibiofilm activity of *T. zygis* was observed on 62.06% of the isolates (18 strains). The percentage of biofilm inhibition ranged from 11.67 to 91.48%. Moreover, three low-grade positive isolates (10.34%) were changed to biofilm negative, and four highly positive isolates (13.79%) become low-grade positive.

The outcomes of this study indicated that *O. majorana* EO had the greatest antibiofilm activity against MRSA isolates succeeded by *R. officinalis* and *T. zygis*.

Person correlation (r) indicated a non-significant correlation between MIC and antibiofilm of EOs (*p* > 0.05). ANOVA test showed a non-significant effect of the specimen on biofilm inhibition (*p* > 0.05).

### 2.6. Biofilm Eradication Activity of Essentials Oils

The results of biofilm eradication activities of EOs are shown in Table 5. The same isolates selected for biofilm inhibitory investigation were used. *O. majorana*, *T. zygis*, and *R. officinalis* EOs showed eradication activities on 41.37% (12 strains) of the MRSA isolates independently of the specimen, including the reference strain. The highest percentage of eradication was recorded with *O. majorana*. The percentage of eradication ranged from 18.31 to 98.01%, from 12.65 to 94.39%, and from 13.45 to 92.69%, respectively, for *O. majorana*, *T. zygis*, and *R. officinalis* EOs.

Based on biofilm categories and under *O. majorana* EO, five isolates (17.24%) were changed from highly positive to low-grade positive. Furthermore, a low-grade positive (isolate number 6) and a highly positive (isolate number 14) strains became biofilm negative. For *T. zygis*, only isolates numbers 5 and 6 were changed from highly positive and low-grade positive respectively to biofilm negative after treatment, while, *R. officinalis* caused modifications on three isolates. Among them, a low-grade positive (isolate number 28) was changed to biofilm negative, and two highly positive (isolates number 5 and 10) became low-grade positive.

Person correlation (r) indicated a non-significant correlation between MIC and biofilm eradication of EOs (*p* > 0.05). ANOVA test showed a non-significant effect of the specimen on biofilm eradication (*p* > 0.05).

## 3. Discussion

Infection caused by MRSA is considered a major public health threat in many countries and MRSA remains the principal cause of hospital and community-acquired infections [21]. This bacterium is accountable for numerous infections related to remarkable morbidity and mortality [22], such as bacteremia, pneumonia, and skin, soft tissue, surgical site, and urinary tract infections [23,24]. This study was conducted on 30 clinical MRSA isolates and results showed variability in the prevalence of the isolates. Indeed, most of the strains (40%) were recovered from SST followed by nasal, burn, SSI, and blood. Akanbi et al. [25] showed that the majority of MRSA strains were isolated from blood, wound, and urine specimens. In addition, Ghebremedhin et al. [26] demonstrated that MRSA was most found in surgical wound infections, succeeded by eye swabs, skin and soft tissue infections.

The ability of MRSA to develop resistance to every antibiotic to which it is exposed makes it a problem to human health [27]. Thereby the development of novel compounds is of great importance. Medicinal plant EOs have been largely used as a natural medicine to combat bacteria, fungi, viruses, and other pathogens [28]. Until now, about 3000 EOs are known, among them 300 are important for industries such as pharmaceutical, food, agronomic, cosmetics, and fragrance. In this work, we investigated the potential antibacterial activities of *O. majorana*, *T. zygis*, and *R. officinalis* medicinal plant EOs against MRSA clinical isolates by disc diffusion, MIC, and MBC techniques. The highest antibacterial activity was observed with *T. zygis*, followed by *O. majorana*, and *R. officinalis* EOs. This result is in agreement with the study of Lagha et al. [13], who showed that *T. zygis* possessed the strongest antimicrobial effect against uropathogenic *E. coli* in contrast to *O. majorana*, and *R. officinalis* EOs. According to biochemical composition, the greater effect of *T. zygis* is owing to linalool (39.7%), which has a strong effect against bacteria and fungi [29]. Regarding *O. majorana* EO, the antibacterial activity can be attributed to the monoterpene alcohol, terpinen-4-ol, as a major compound (25.9%) [30], which was found to be effective against MRSA [31]. According to Cordeiro et al. [32], terpinen-4-ol has a powerful antibacterial effect against *S. aureus*. This compound functions as a bactericidal by obstructing the synthesis of the cell wall. Moreover, other main components such as terpinen-4-ol (11.7%), β-myrcene (8.6%) and γ-terpinene (7.6%) are present in *T. zygis* in addition to linalool (10.9%), γ-terpinene (16.9%) and α-terpinene (7.7%) present in *O. majorana* may enhance the antibacterial effect of these oils. Concerning *R. officinalis* EO, which showed the lowest activity against MRSA isolates, its antibacterial activity may be related to α-pinene (37.7%) as a major constituent. The study of Leite et al. [33] showed antibacterial activity of α-pinene against *S. aureus* and *S. epidermidis*. Other reports [34,35] revealed the antibacterial activity of some EOs against Gram-negative and Gram-positive bacteria when α-Pinene is the major constituent. However, Utegenova et al. [36] demonstrated that α-pinene had low activity against MRSA, indicating that other components were probably responsible. Among them, in this study, 1,8-cineole (4.7%) altered the structure of *E. coli*, *S. enteritidis*, and *S. aureus* [37]. The antibacterial activity of *R. officinalis* could be attributed to the synergistic effect of camphor (5.2%), verbenone (5.4%), and borneol (5.5%) in addition to α-pinene and 1,8-cineole [38]. In general, whole essential oils have an important antimicrobial effect compared to the major compounds individually or collectively. This suggests that minor constituents are essential and may have a synergistic antibacterial effect [10].

MRSA were tested for their capacities to produce biofilm on polystyrene microplates and results indicated that 40% of the strains were highly positive biofilm and 56.66% were low-grade positive. Out of the 30 isolates, only one strain was biofilm negative, which indicates the high potentiality of the isolates to form biofilm. Biofilm, as a virulence factor that favors the chronicity of *S*. *aureus* infections, is accountable for more 65% of nosocomial infections and 80% of microbial infections [5]. Biofilm is related to various staphylococcal diseases, such as skin and soft tissue infections, nasal colonization, endocarditis and urinary tract infections [39]. Further, biofilm becomes a serious threat in the urology field due to its responsibility for the long persistence of bacteria in the genitourinary tract [40]. The high ability of the investigated isolates to form biofilm confirms the fact that *S. aureus* is the leading species in biofilm-associated infection.

Biofilm has been associated with medical devices and its treatment is becoming increasingly difficult due to the resistance to antibiotics and the immune system in addition to the spread of infection [39]. Thereby, the development of new therapeutic strategies, such as EOs, to inhibit or eradicate biofilm is great of interest. In this work, biofilm inhibitory activity of EOs showed that *O. majorana* had the highest antibiofilm activity (antibiofilm effect on 89.66% of the isolates) against MRSA isolates followed by *R. officinalis* and *T. zygis* that demonstrated activity on 79.31 and 62.06% of the isolates, respectively. EOs also showed a strong potential to inhibit biofilm, with percentage of inhibition ranging from 10.29 to 95.91%, from 10.20 to 95.65%, and from 11.67 to 91.48%, respectively for *O. majorana*, *R. officinalis*, and *T. zygis*. Based on our results, the oil with the highest growth inhibition activity was different from the oil with the highest biofilm inhibition effect, which indicates that the components involved in growth inhibition were different from those associated with biofilm inhibition. According to biochemical specificity, terpinen-4-ol present in *O. majorana* as major compound, has more inhibition of the biofilm formation process by MRSA isolates compared to α-pinene and linalool present, as the main components, in *R. officinalis* and *T. zygis* EOs, respectively.

This finding corroborates the recent data of Cordeiro et al. [32] showing that the strongest antibiofilm activity of terpinen-4-ol was against *S. aureus*. Other studies have also demonstrated that this compound possesses an excellent potential against biofilm formed by some pathogenic bacteria like *Pseudomonas aeruginosa* [41], *Streptococcus mutans*, *Lactobacillus acidophilus* [42], *Porphyromonas gingivalis*, and Fusobacterium nucleatum [43]. Biofilm inhibition properties of *O. majorana*, *R. officinalis*, and *T. zygis* EOs against MRSA suggest that the addition of these oils before biofilm formation eliminates planktonic cells and may reduce the polystyrene surface adherence, which becomes less susceptible to cell adhesion. Additionally, the modification of MRSA surface proteins caused by their interactions with oils inhibits the adhesion of this bacterium to the polystyrene surface, which is the initial attachment phase [44].

Preformed biofilms are difficult to eradicate by conventional antibiotic therapy. However, in the present study, *O. majorana*, *T. zygis*, and *R. officinalis* EOs showed high biofilm eradication activities on 41.37% of the MRSA isolates. *O. majorana* EO had the strongest effect, with the percentage of eradication going up to 98.01%, and seven isolates were changed their biofilm phenotype. It seems that the monoterpenoid terpinen-4-ol has an excellent potential to eradicate mature biofilm than α-pinene and linalool. The activity of these oils on mature biofilms was lower than their capacity to inhibit the formation of biofilms. This can be explained by the fact that the major constituents in these oils have an effect on the biofilm formation process more than on mature biofilm. This is in agreement with the report of Cordeiro et al. [32], showing that terpinen-4-ol is more efficient in inhibiting the formation of *S. aureus* biofilms than in breaking or eliminating mature biofilms. Many EOs, such as tea tree [45], eucalyptus [46], and cinnamon oil [47] have shown their effective ability to remove biofilm. Moreover, *R. officinalis* EO has reduced the quantity of *S. aureus* biofilm to 60.76% [48]. In general, EOs diffuse through polysaccharide matrix of the preformed biofilm and destabilize it because of higher intrinsic antimicrobial activities [44].

## 4. Materials and Methods

### 4.1. Bacterial Strains

Thirty clinical MRSA isolates were collected from King Abdulaziz Specialist Hospital, Taif, Saudi Arabia. The isolates were identified as *S. aureus*, as described previously [49]. The methicillin resistance phenotype was performed by the Vitek 2 system (bioMérieux, Durham, North CA, USA) in accordance with the British Society for Antimicrobial Chemotherapy (BSAC). Each isolate was considered as methicillin-resistant when the minimum inhibitory concentration (MIC) breakpoint of oxacillin was >2 mg/L and cefoxitin >4 mg/L. [50]. *S. aureus* ATCC 25922 was used as control.

### 4.2. Medicinal Plants Essential Oils

Three commercial EOs extracted from medicinal plants were investigated. These EOs were bought from Laboratoires OMEGA Pharma (Groupe Perrigo)—Phytosun Arôms (Châtillon, France) and kept at 4 °C in dark glass bottles till used. These oils were extracted from twigs of *R. officinalis* L. (M14302), and from the aerial parts of flowering stage of *T. zygis* L. subsp. *zygis* (M13184) and *O. majorana* L. (74K100C6). These EOs were carefully chosen for their antibacterial and/or antibiofilm actions, as stated previously [11,12,13] and their usage in traditional medicine.

### 4.3. Gas Chromatography—Mass Spectrometry Analysis

The GC-MS analysis was performed as described previously [51].

### 4.4. Antibacterial Activity of Essential Oils

#### 4.4.1. Disc Diffusion

The agar disc diffusion method was used to evaluate the antibacterial activities of the EOs [52]. Briefly, an overnight cultures of MRSA cells grown at 37 °C were diluted to a density of 0.5 McFarland standards turbidity (DENSIMAT, Bio-merieux, Marcy l’Etoile, France) and were streaked onto Mueller–Hinton agar (MHA) plates using a sterile swab. A sterile filter disc (diameter 6 mm) was placed and then was impregnated by *R. officinalis*, *T. zygis*, and *O. majorana* EOs (10 μL /disc). The plates were maintained at 4 °C for 2 h and then incubated at 37 °C for 24 h. After incubation, the antibacterial activity was evaluated by determining the zone of growth inhibition throughout the discs.

Inhibitory action was categorized according to the zone of inhibition (ZI) as described previously [13,14,15,16,17,18,19,20,21,22,23,24,25,26,27,28,29,30,31,32,33,34,35,36,37,38,39,40,41,42,43,44,45,46,47,48,49,50,51,52,53]. The experiment was performed in triplicate, and the mean diameter of the inhibition zone was documented.

#### 4.4.2. Minimum Inhibitory Concentration and Minimum Bactericidal Concentration

The minimal inhibition concentration (MIC) and the minimal bactericidal concentration (MBC) were assessed in triplicates on 96-well microtiter plates (Nunc, Roskilde, Denmark) as described previously [54]. A bacterial suspension at a density of 0.5 McFarland standards turbidity was prepared from an overnight culture. Then, a serial two-fold dilution for each EOs (50 mg/mL stock solution) was made in 5 mL of nutrient broth with concentration ranged from 0.012–50 mg/mL.

Each well of the 96-well plates contains 95 μL of nutrient broth and 5 μL of the bacterial inoculum. A 100 μL aliquot from the stock solutions of each EO was added into the first well. Then, 100 μL from the serial dilutions were transferred into the consecutive well. The negative control well contains 195 μL of nutrient broth without EO and 5 μL of the bacterial inoculum. The final volume in each well was 200 μL, and the plates were incubated at 37 °C for 18–24 h.

The MIC was defined as the lowest concentration of the EO at which the MRSA cells growth is inhibited. The MBC was determined by subculturing 20 μL from clear wells of the MIC test on MHA. MBC was defined as the lowest concentration of EOs, required to kill ≥99.9% of the initial bacterial inoculum [55].

### 4.5. Biofilm Formation

Biofilm formation by MRSA isolates was determined using crystal violet assay on U-bottomed 96-well microtiter plates, as detailed previously [56]. Each MRSA strain was tested three times. Wells with sterile TSB only were worked as controls. The optical density of each well was measured at 570 nm (OD570) using an automated Multiskan reader (GIO. DE VITA E C, Rome, Italy). Biofilm formation was interpreted as highly positive (OD570 ≥ 1), low-grade positive (0.1 ≤ OD570 < 1), or negative (OD570 < 0.1).

### 4.6. Biofilm Inhibition

EOs were tested for their potential to prevent biofilm formation by MRSA isolates. For the experiment, 100 µL of the EOs emulsified in TSB supplement with 2% glucose were put in the U-bottomed 96-well microtiter plate, including 100 μL of bacterial suspensions (10^8^ CFU/mL) in each well. The final concentrations of the EOs were equal to MIC, and the final volume was 200 μL per well. The analyzes were performed three times. After incubation of microplates at 37 °C for 24 h, the formed biofilm was measured by crystal violet as described previously [56]. For the Controls wells, the inoculums volume and EOs were replaced by TSB and sterile water, respectively. Inhibition of biofilm was determined from the formula described by Jadhav et al. [57].
$$\%\;\mathrm{Inhibition}\;=\;100-(\frac{\mathrm{OD}570\;\mathrm{sample}}{\mathrm{OD}570\;\mathrm{control}}\;\times \;100)$$

### 4.7. Biofilm Eradication

In order to eradicate the preformed biofilm at the maturation stage (48 h biofilms), the plates were incubated for 48 h, the medium was changed after 24 h, and EOs were added at the same concentrations and at the last 24 h. Biofilms formed by bacteria that did not undergo any treatment were used as controls. Experimentally, the plates were incubated for 24 h at 37 °C to allow for biofilm attachment and growth. The following day, the non-adhered cells were removed from each well, and the adhered biofilm was rinsed two times with PBS. Then, 200 µL of TSB (2% glucose) with final concentrations of the EOs equivalent to MIC was added, and the plates were incubated for 24 h [44]. The biofilm was assessed by crystal violet, and eradication of biofilm was calculated as described in Section 4.6. The experiment was carried out in triplicate.

### 4.8. Statistical Analysis

Statistical analysis was performed using analysis of variance (ANOVA). Pearson’s simple linear correlation coefficient (r) and their significance (*p*) were assessed using IBM SPSS (v20).

## 5. Conclusions

The outcomes of this study support the medical application of *O. majorana*, *T. zygis*, and *R. officinalis* EOs for the prevention and/or treatment of MRSA infections and diseases as an alternative to or combined with antibiotics. These EOs, provided from Laboratoires OMEGA Pharma–Phytosun Arôms (Châtillon, France), are used orally and in high concentrations (doses), corroborating their non-toxic effect. Generally, the therapeutic application of EOs is limited by their solubility, skin-sensitization synonymous allergic contact dermatitis, and their physicochemical stability due to the volatile components and the conversion of components by cyclization, isomerization, oxidation, or dehydrogenation reactions. Further adequate in vitro testing or in vivo preclinical experiments are warranted to establish safety, efficacy, potential adverse effects, and interaction with other drugs of *O. majorana*, *T. zygis*, and *R. officinalis* EOs before including them in clinical practice.

## Figures and Tables

**Figure 1 pharmaceuticals-13-00369-f001:**
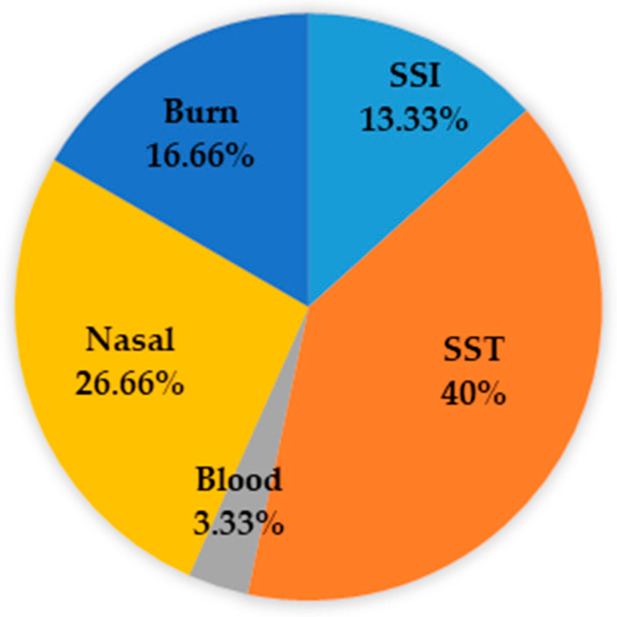
Distribution of MRSA isolates.

**Table 1 pharmaceuticals-13-00369-t001:** Chemical composition of the essential oils.

Components	*O. majorana* (%) [13]	*T. zygis* (%) [13]	*R. officinalis* (%)
α-Pinene	0.46	3.6	37.7
Sabinene	8	0.84	
β-Pinene	1.4	0.33	1.1
β-Myrcene	1.1	8.6	
α-phellandrene	0.30	0.48	
Limonene	3.5	2.6	4.1
Terpinen-4-ol	25.9	11.7	
Bornyl acetate		0.07	9.1
β-Caryophyllene	2.3	1.6	
α-Thujene	0.33	0.21	
Camphene	0.03	0.74	7.3
α-Terpinene	7.7	4.2	
*p*-Cymene	3.4	2.2	
1,8-Cineole	0.15		4.7
γ-Terpinene	16.9	7.6	
Terpinolene	1.7	2	
Linalool	10.9	39.7	1.8
Borneol		1.9	5.5
α-Terpineol	2.5	1.7	
Camphor		0.22	5.2
α-Humulene	0.05		
*cis* and *trans*-thujan-4-ol	2.2–2.3	0.88–2.2	
*cis* and *trans* piperitol	0.13–0.18	0.13–0.08	
Linalyl acetate	7	0.5	
Carvacrol	0.03	0.08	
Thymol	0.05	0.52	
Bicyclogermacrene	0.41	0.16	
*Cis* and *trans*-p-menth-2-en-1-ol	0.59–0.32	0.37–0.25	
Caryophyllene oxide	0.04		
Ocimene	0.07		
Spathulenol	0.01		
*cis*-Dihydrocarvone		0.17	
trans-Dihydrocarvone		0.2	
Verbenone			5.4

**Table 2 pharmaceuticals-13-00369-t002:** Antimicrobial effect of EOs against MRSA isolates using disc diffusion.

Essential Oils	Inhibitory Action				
	Strong *n* (%)	Complete *n* (%)	Partial *n* (%)	Slight *n* (%)	No Action *n* (%)
*O. majorana*	16 (53.33%)	11 (36.66%)	3 (10%)		
*T. zygis*	24 (80%)	6 (20%)			
*R. officinalis*	5 (16.66%)	5 (16.66%)	6 (20%)	14 (46.66%)	

*n*: number of isolates.

**Table 3 pharmaceuticals-13-00369-t003:** Biofilm formation ability of MRSA isolates on polystyrene surface.

Isolates	Specimen	OD570 ± SD	Biofilm Formation
1	Burn	0.24 ± 0.026	low-grade positive
2	Blood	2.609 ± 0.088	highly positive
3	SST	3.635 ± 0.052	highly positive
4	SST	1.437 ± 0.074	highly positive
5	Nasal	1.175 ± 0.03	highly positive
6	SSI	0.147 ± 0.028	low-grade positive
7	SST	0.135 ± 0.031	low-grade positive
8	Burn	1.971 ± 0.049	highly positive
9	Nasal	0.19 ± 0.079	low-grade positive
10	Nasal	1.378 ± 0.06	highly positive
11	SST	0.194 ± 0.075	low-grade positive
12	SST	1.554 ± 0.086	highly positive
13	SST	0.305 ± 0.021	low-grade positive
14	Nasal	2.157 ± 0.071	highly positive
15	Nasal	0.045 ± 0.007	Negative
16	SST	0.198 ± 0.078	low-grade positive
17	SSI	0.87 ± 0.023	low-grade positive
18	Nasal	0.221 ± 0.048	low-grade positive
19	Burn	0.428 ± 0.068	low-grade positive
20	Nasal	0.745 ± 0.018	low-grade positive
21	Nasal	0.319 ± 0.012	low-grade positive
22	SST	0.233 ± 0.087	low-grade positive
23	SST	0.788 ± 0.027	low-grade positive
24	SSI	0.642 ± 0.028	low-grade positive
25	SST	1.836 ± 0.038	highly positive
26	Burn	2.696 ± 0.054	highly positive
27	SST	0.418 ± 0.056	low-grade positive
28	Burn	0.438 ± 0.067	low-grade positive
29	SSI	0.113 ± 0.045	low-grade positive
30	SST	2.308 ± 0.039	highly positive
ATCC 25923		3.36 ± 0.098	highly positive

SST: skin and soft tissue; SSI: surgical site infection.

**Table 4 pharmaceuticals-13-00369-t004:** Biofilm inhibition activity of EOs against MRSA isolates.

Isolates	Control OD570 ± SD	*O. majorana* OD570 ± SD	Inhibition (%)	*T. zygis* OD570 ± SD	Inhibition (%)	*R. officinalis* OD570 ± SD	Inhibition (%)
1	0.24 ± 0.026	0.061 ± 0.004 *	74.58	0.112 ± 0.015	53.33	0.238 ± 0.028	0
2	2.609 ± 0.088	2.603 ± 0.093	0	2.61 ± 0.019	0	2.595 ± 0.098	0
3	3.635 ± 0.052	2.701 ± 0.082	25.69	3.658 ± 0.01	0	2.744 ± 0.066	24.51
4	1.437 ± 0.074	0.131 ± 0.038 **	90.88	0.965 ± 0.022 **	32.84	0.728 ± 0.084 **	49.33
5	1.175 ± 0.03	0.048 ± 0.006 ***	95.91	0.1 ± 0.038 **	91.48	0.051 ± 0.004 ***	95.65
6	0.147 ± 0.028	0.03 ± 0.008 *	79.59	0.124 ± 0.043	15.64	0.132 ± 0.023	10.20
7	0.135 ± 0.031	0.114 ± 0.023	15.55	0.136 ± 0.073	0	0.134 ± 0.011	0
8	1.971 ± 0.049	0.105 ± 0.018 **	94.67	0.239 ± 0.088 **	87.87	0.346 ± 0.018 **	82.44
9	0.19 ± 0.079	0.049 ± 0.009 *	74.21	0.138 ± 0.043	27.36	0.146 ± 0.093	23.15
10	1.378 ± 0.06	1.387 ± 0.038	0	1.369 ± 0.054	0	0.828 ± 0.082 **	39.91
11	0.194 ± 0.075	0.02 ± 0.005 *	89.69	0.142 ± 0.058	26.80	0.111 ± 0.044	42.78
12	1.554 ± 0.086	0.162 ± 0.077 **	89.57	0.22 ± 0.077 **	85.84	0.17 ± 0.023 **	89.06
13	0.305 ± 0.021	0.027 ± 0.006 *	91.14	0.303 ± 0.032	0	0.301 ± 0.069	0
14	2.157 ± 0.071	1.935 ± 0.014	10.29	2.154 ± 0.04	0	1.724 ± 0.092	20.07
16	0.198 ± 0.078	0.038 ± 0.009 *	80.80	0.072 ± 0.008 *	63.63	0.053 ± 0.006 *	73.23
17	0.87 ± 0.023	0.043 ± 0.017 *	95.05	0.124 ± 0.089	85.74	0.16 ± 0.026	81.60
18	0.221 ± 0.048	0.105 ± 0.028	52.48	0.142 ± 0.037	35.74	0.122 ± 0.038	44.79
19	0.428 ± 0.068	0.053 ± 0.039	87.61	0.095 ± 0.002 *	77.80	0.226 ± 0.077	47.19
20	0.745 ± 0.018	0.055 ± 0.033 *	92.61	0.11 ± 0.082	85.23	0.562 ± 0.065	24.56
21	0.319 ± 0.012	0.072 ± 0.013 *	77.42	0.317 ± 0.075	0	0.085 ± 0.004 *	73.35
22	0.233 ± 0.087	0.231 ± 0.032	0	0.131 ± 0.012	43.77	0.089 ± 0.003 *	61.80
23	0.788 ± 0.027	0.554 ± 0.065	29.69	0.696 ± 0.028	11.67	0.465 ± 0.073	40.98
24	0.642 ± 0.028	0.192 ± 0.013	70.09	0.639 ± 0.087	0	0.241 ± 0.053	62.461
25	1.836 ± 0.038	1.325 ± 0.078	27.83	1.84 ± 0.042	0	1.821 ± 0.096	0
26	2.696 ± 0.054	2.196 ± 0.04	18.54	2.206 ± 0.07	18.17	2.012 ± 0.014	25.37
27	0.418 ± 0.056	0.254 ± 0.068	39.23	0.415 ± 0.069	0	0.065 ± 0.008 *	84.44
28	0.438 ± 0.067	0.034 ± 0.005 *	92.23	0.217 ± 0.016	50.45	0.069 ± 0.002 *	84.24
29	0.113 ± 0.045	0.11 ± 0.022	0	0.016 ± 0.006 *	85.84	0.114 ± 0.032	0
30	2.308 ± 0.039	1.533 ± 0.055	33.57	2.306 ± 0.086	0	0.907 ± 0.048 **	60.70
ATCC 25922	3.36 ± 0.098	1.838 ± 0.066	45.29	3.352 ± 0.014	0	3.345 ± 0.029	0

* Isolates changed from low-grade positive to biofilm negative after treatment with EOs. ** Isolates changed from highly positive to low-grade positive after treatment with EOs. *** Isolates changed from highly positive to biofilm negative after treatment with EOs.

**Table 5 pharmaceuticals-13-00369-t005:** Biofilm eradication activity of EOs against MRSA isolates.

Isolates	Control OD570 ± SD	*O. majorana* OD570 ± SD	Eradication (%)	*T. zygis* OD570 ± SD	Eradication (%)	*R. officinalis* OD570 ± SD	Eradication (%)
1	0.418 ± 0.024	0.417 ± 0.015	0	0.415 ± 0.019	0	0.413 ± 0.021	0
2	3.13 ± 0.096	1.754 ± 0.055	43.96	1.626 ± 0.06	48.05	2.709 ± 0.084	13.45
3	4.362 ± 0.086	2.92 ± 0.071	33.05	3.468 ± 0.091	20.49	2.15 ± 0.079	50.71
4	1.939 ± 0.078	1.941 ± 0.069	0	1.931 ± 0.074	0	1.115 ± 0.063	42.49
5	1.41 ± 0.088	0.237 ± 0.025 **	83.19	0.079 ± 0.008 ***	94.39	0.103 ± 0.011 **	92.69
6	0.176 ± 0.016	0.078 ± 0.005 *	55.68	0.061 ± 0.004 *	65.34	0.131 ± 0.018	25.56
7	0.189 ± 0.012	0.185 ± 0.013	0	0.191 ± 0.015	0	0.186 ± 0.015	0
8	2.465 ± 0.083	0.37 ± 0.045 **	84.98	2.153 ± 0.079	12.65	1.143 ± 0.059	53.63
9	0.304 ± 0.056	0.308 ± 0.021	0	0.309 ± 0.022	0	0.302 ± 0.024	0
10	1.722 ± 0.066	0.386 ± 0.027 **	77.58	1.074 ± 0.055	37.63	0.174 ± 0.014 **	89.89
11	0.269 ± 0.013	0.262 ± 0.016	0	0.265 ± 0.025	0	0.271 ± 0.02	0
12	2.334 ± 0.073	0.687 ± 0.032 **	70.56	1.356 ± 0.058	41.90	2.328 ± 0.026	0
13	0.433 ± 0.026	0.431 ± 0.026	0	0.436 ± 0.031	0	0.43 ± 0.032	0
14	2.617 ± 0.078	0.052 ± 0.003 ***	98.01	2.614 ± 0.082	0	2.609 ± 0.079	0
16	0.286 ± 0.034	0.287 ± 0.02	0	0.281 ± 0.015	0	0.288 ± 0.019	0
17	1.249 ± 0.028	1.247 ± 0.063	0	1.241 ± 0.068	0	1.244 ± 0.064	0
18	0.298 ± 0.011	0.292 ± 0.018	0	0.294 ± 0.016	0	0.296 ± 0.023	0
19	0.676 ± 0.026	0.679 ± 0.039	0	0.671 ± 0.029	0	0.674 ± 0.038	0
20	0.894 ± 0.045	0.448 ± 0.028	49.88	0.613 ± 0.036	31.43	0.584 ± 0.034	34.67
21	0.516 ± 0.068	0.514 ± 0.04	0	0.513 ± 0.031	0	0.517 ± 0.041	0
22	0.319 ± 0.022	0.313 ± 0.024	0	0.317 ± 0.024	0	0.314 ± 0.021	0
23	1.194 ± 0.039	1.19 ± 0.052	0	1.195 ± 0.057	0	1.192 ± 0.051	0
24	0.808 ± 0.027	0.801 ± 0.048	0	0.805 ± 0.044	0	0.81 ± 0.046	0
25	2.249 ± 0.069	2.245 ± 0.069	0	1.824 ± 0.072	18.89	1.806 ± 0.058	19.69
26	3.396 ± 0.094	1.674 ± 0.049	50.70	1.785 ± 0.058	47.43	2.025 ± 0.06	40.37
27	0.568 ± 0.021	0.561 ± 0.034	0	0.567 ± 0.039	0	0.564 ± 0.039	0
28	0.535 ± 0.019	0.437 ± 0.025	18.31	0.337 ± 0.03	37	0.081 ± 0.006 *	84.85
29	0.133 ± 0.011	0.134 ± 0.019	0	0.136 ± 0.026	0	0.132 ± 0.011	0
30	2.989 ± 0.083	0.348 ± 0.023 **	88.35	1.342 ± 0.056	55.10	1.756 ± 0.053	41.25
ATCC 25922	4.32 ± 0.098	2.796 ± 0.085	35.27	1.972 ± 0.068	54.35	2.577 ± 0.087	40.34

* Isolates changed from low-grade positive to biofilm negative after treatment with EOs. ** Isolates changed from highly positive to low-grade positive after treatment with EOs. *** Isolates changed from highly positive to biofilm negative after treatment with EOs.

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
