# Peer review of "Biofilm Inhibition and Eradication Properties of Medicinal Plant Essential Oils against Methicillin-Resistant Staphylococcus aureus Clinical Isolates"

_pharmaceuticals, 2020, doi:10.3390/ph13110369_

Round 1

Reviewer 1 Report

In this report, Abdallah et al raised the important question: “Can plant essential oils serve as novel antimicrobial agents against MRSA isolates”. Tested oils have all previously been shown to exhibit anti-S. aureus capacities, and so potential novelty of the enclosed findings are restricted to the use of MRSA. Yet, to corroborate Methicillin-resistance (or the level hereof), it is imperative that the authors include methicillin as control in their assays. Unfortunately, this has not been done, and so the current manuscript is merely a replication (even the same methods) as their own ref 11, who already in 2015 demonstrated anti-S. aureus capabilities of the tested oils against isolates from poultry meat.

Minor comments:

While it is obvious the authors did a pertinent job in writing and conveying a clear message (although essential controls are missing), the MS suffers from a number of minor grammatical errors. Please find below a few comments to this end. The number of minor mistakes was too overwhelming to being corrected by this reviewer, so the below comments is only a snapshot of the actual amount of grammatical errors. The MS would therefore benefit from being edited by a native speaker before potential publication (after inclusion of appropriate controls).

Abstract:

Line 14: Insert a comma after Biofilms, or alternatively delete the comma after therapy, line 15.

Line 16: Change ‘finding’ to ‘report’.

Line 24-25: Change ‘has’ to ‘had’ and delete ‘compared to R. offoconalis and T. zygis’.

Introduction:

Line 32: Either ‘among the most common causes of nosocomial infections’ or ‘the most common causes of nosocomial infections’. Causes can only be plural if there are several.

Line 36: delete ‘an’ before ‘appropriate’.

Author Response

University of Monastir, Tunisia 
Higher Institute of Biotechnology of Monastir, Tunisia 

Taif University, Saudi Arabia

Department of Biology, College of Sciences

November 1st, 2020 

Dear Reviewer 

Please find enclosed our revised manuscript entitled “Biofilm inhibition and eradication properties of medicinal plant essential oils against Methicillin-resistant Staphylococcus aureus clinical isolates”. This article has been submitted for publication in “Pharmaceuticals”. 

Thank you for the reviewer comments pertaining to our manuscript. We are happy to submit a revised version (where corrections were labeled with a blue color in the manuscript) and here is a point to point reply to the comments

In this report, Abdallah et al raised the important question: “Can plant essential oils serve as novel antimicrobial agents against MRSA isolates”. Tested oils have all previously been shown to exhibit anti-S. aureus capacities, and so potential novelty of the enclosed findings are restricted to the use of MRSA. Yet, to corroborate Methicillin-resistance (or the level hereof), it is imperative that the authors include methicillin as control in their assays. Unfortunately, this has not been done, and so the current manuscript is merely a replication (even the same methods) as their own ref 11, who already in 2015 demonstrated anti-S. aureus capabilities of the tested oils against isolates from poultry meat.

-We thank the reviewer for highlighting these points. The use of methicillin or any antibiotic in such study as control is to evaluate the antibacterial effect of any investigated compound. However, in this study we used other method independently of methicillin in order to evaluate the antibacterial effect of the oils (reference 55), which allows us to classify the effect of oils in 5 categories based on the diameter of inhibition.

-In the reference 11, authors studied only the antibacterial effect of Origanum vulgare and Origanum majorana against S. aureus isolates from poultry meat. However, in our work we studied the antibacterial, biofilm inhibition and biofilm eradication activities of Rosmarinus officinalis, Thymus zygis that not investigated in reference 11, in addition to Origanum majorana against clinical MRSA. which shows that our study is different and much more advanced than reference 11 in term of effect of oils against S. aureus.

-Authors of reference 11 conclude the possible use of oils in food preservation, however, our work supports the medical application of O. majorana, T. zygis, and R. officinalis EOs for the prevention and/or treatment of MRSA infections and diseases, in addition to the difference between the two works in term of oils used, strains, and their origin.

- the methods used for the antibacterial effect such as disc diffusion, MIC and MBC are standard techniques and are the first step to evaluate the effect of any compound against bacteria.

Minor comments:

While it is obvious the authors did a pertinent job in writing and conveying a clear message (although essential controls are missing), the MS suffers from a number of minor grammatical errors. Please find below a few comments to this end. The number of minor mistakes was too overwhelming to being corrected by this reviewer, so the below comments is only a snapshot of the actual amount of grammatical errors. The MS would therefore benefit from being edited by a native speaker before potential publication (after inclusion of appropriate controls).

We thank the reviewer for this comment. Accordingly, the English language was checked through the manuscript.

Abstract:

Line 14: Insert a comma after Biofilms, or alternatively delete the comma after therapy, line 15.

Line 16: Change ‘finding’ to ‘report’.

Line 24-25: Change ‘has’ to ‘had’ and delete ‘compared to R. offoconalis and T. zygis’.

 We thank the reviewer for bringing our attention to these points. Accordingly, these modifications were changed in the text.

Introduction:

Line 32: Either ‘among the most common causes of nosocomial infections’ or ‘the most common causes of nosocomial infections’. Causes can only be plural if there are several.

Line 36: delete ‘an’ before ‘appropriate’.

 We thank the reviewer for highlighting these points. Accordingly, these modifications were changed in the text.

We hope that these responses are satisfactory and that you will now find this manuscript suitable for publication in “Pharmaceuticals”.

Looking forward to hearing from you,

Yours faithfully,

Reviewer 2 Report

In the present manuscript, authors aimed to study the biofilm inhibition and eradication properties of Origanum majorana, Rosmarinus officinalis and Thymus zygis medicinal plant essential oils against Methicillin-resistant Staphylococcus aureus clinical isolates.

The topic of the work is of great interest, but there are some issues and some crucial points that need consideration. At first, the English style must be improved throughout the text and some grammar, editing and typing errors have to be corrected. I suggest minor revision of the manuscript before to assess it for publication in Pharmaceuticals. In particular:

Introduction

Page 1, lines 33-34: “MRSA causes several diseases ranging from skin infections to serious invasive infections such as skin and soft tissues infections pneumonia…”. The skin infections are repeated twice. Please revised this sentence.

Page 2, lines 49-50: “Rosmarinus officinalis, Thymus zygis and Origanum majorana EOs have been reported for their antibacterial…”. Please define correctly the plant material by reporting the Latin binomial (genus, species), variety, author, family and part of the plant used to obtain the essential oil.

Results

Page 4, lines 86-87: “Moreover, according on the high percentage of anti-MRSA activity, T. zygis and O. majorana EOs have a strong inhibitory action on 80.64% and 54.83% of the strains respectively”. I did not understand to what the percentage are referred to. Indeed, the value reported in the Table 2 are different and are already reported in the previous sentence. Please clarify this point.

Page 4, lines 96-98: “Concerning T. zygis EO, The MICs were ranged from 0.39 mg/ml to 0.78 mg/ml, while the MBC value was 3.125 mg/ml. However, The MIC of R. officinalis ranged from 0.78 mg/ml to 1.56 mg/ml and the MBC was 12.5 mg/ml).” Please, correct editing and typing errors.

Discussion

Page 8, lines 153-155: “This bacterium is accountable for numerous infections related with remarkable morbidity and mortality [22] such as bacteremia, pneumonia, skin and soft tissue, surgical site, urinary tract [23–24].”. Please replaced with “This bacterium is accountable for numerous infections related with remarkable morbidity and mortality [22] such as bacteremia, pneumonia, and skin, soft tissue, surgical site, and urinary tract infections [23–24].”.

Page 9, line 236: “…R. officinalis EO has reduced the quantity of by S. aureus biofilm to 60.76%...”. Please remove “by” before “S. aureus biofilm”.

Conclusions

This section should be rewritten and amplified taking into account the limits of essential oil therapeutic application, such as limited solubility, poor physicochemical stability and potential skin-sensitization (e.g. contact dermatitis). Moreover, the high doses tested in present study should be justify considering their potential application in humans.

Author Response

University of Monastir, Tunisia 
Higher Institute of Biotechnology of Monastir, Tunisia 

Taif University, Saudi Arabia

Department of Biology, College of Sciences

November 1st, 2020 
Dear Reviewer 

Please find enclosed our revised manuscript entitled “Biofilm inhibition and eradication properties of medicinal plant essential oils against Methicillin-resistant Staphylococcus aureus clinical isolates”. This article has been submitted for publication in “Pharmaceuticals”. 

Thank you for the reviewer comments pertaining to our manuscript. We are happy to submit a revised version (where corrections were labeled with a blue color in the manuscript) and here is point to point reply to the comments

The topic of the work is of great interest, but there are some issues and some crucial points that need consideration. At first, the English style must be improved throughout the text and some grammar, editing and typing errors have to be corrected. I suggest minor revision of the manuscript before to assess it for publication in Pharmaceuticals. In particular:

We thank the reviewer for this comment. Accordingly, the English language was checked through the manuscript.

Introduction

Page 1, lines 33-34: “MRSA causes several diseases ranging from skin infections to serious invasive infections such as skin and soft tissues infections pneumonia…”. The skin infections are repeated twice. Please revised this sentence.

We thank the reviewer for bringing our attention to this point. Accordingly, the sentence was revised.

Page 2, lines 49-50: “Rosmarinus officinalis, Thymus zygis and Origanum majorana EOs have been reported for their antibacterial…”. Please define correctly the plant material by reporting the Latin binomial (genus, species), variety, author, family and part of the plant used to obtain the essential oil.

We thank the reviewer for highlighting this point. Accordingly, this modification was changed in the text.

Results

Page 4, lines 86-87: “Moreover, according on the high percentage of anti-MRSA activity, T. zygis and O. majorana EOs have a strong inhibitory action on 80.64% and 54.83% of the strains respectively”. I did not understand to what the percentage are referred to. Indeed, the value reported in the Table 2 are different and are already reported in the previous sentence. Please clarify this point.

We thank the reviewer for bringing our attention to this point. We apologize for this typing errors and the percentages were corrected.

Page 4, lines 96-98: “Concerning T. zygis EO, The MICs were ranged from 0.39 mg/ml to 0.78 mg/ml, while the MBC value was 3.125 mg/ml. However, The MIC of R. officinalis ranged from 0.78 mg/ml to 1.56 mg/ml and the MBC was 12.5 mg/ml).” Please, correct editing and typing errors.

We thank the reviewer for this comment. Accordingly, this modification was changed in the text.

Discussion

Page 8, lines 153-155: “This bacterium is accountable for numerous infections related with remarkable morbidity and mortality [22] such as bacteremia, pneumonia, skin and soft tissue, surgical site, urinary tract [23–24].”. Please replaced with “This bacterium is accountable for numerous infections related with remarkable morbidity and mortality [22] such as bacteremia, pneumonia, and skin, soft tissue, surgical site, and urinary tract infections [23–24].”.

We thank the reviewer for this comment. Accordingly, this modification was changed in the text.

Page 9, line 236: “…R. officinalis EO has reduced the quantity of by S. aureus biofilm to 60.76%...”. Please remove “by” before “S. aureus biofilm”.

We thank the reviewer for this comment. Accordingly, this modification was changed in the text.

Conclusions

This section should be rewritten and amplified taking into account the limits of essential oil therapeutic application, such as limited solubility, poor physicochemical stability and potential skin-sensitization (e.g. contact dermatitis). Moreover, the high doses tested in present study should be justify considering their potential application in humans.

We thank the reviewer for this comment. The conclusion was modified according to your pertinent comments.

These EOs were purchased from “Laboratoires OMEGA Pharma– Phytosun Arôms (https://www.phytosunaroms.com/)”. These oils are already marketed and used orally by humans in high concentrations according to the laboratory which shows their safety.

- O. majorana: 186mg/day

- T. zygis: 281mg/day

- R. officinalis: 200mg/day

We hope that these responses are satisfactory, and that you will now find this manuscript suitable for publication in “Pharmaceuticals”.

Looking forward to hearing from you,

Yours faithfully,

Reviewer 3 Report

Suggestions:

 Check

 Thymus zygis ssp. gracilis or sylvestris or zygis (Boiss) ; Origanum majorana L.; Rosmarinus officinalis L.

Particularly is the so low concentration of Thymol, is it referable to the Thymus zygis subspecies? can authors comment on it?

Introduction

added the following references as n.11 and 12

[11] Rota, MC.; Herrera, A.; Martinez, RM.; Sotomayor, JA.; Jordan MJ. Antimicrobial activity and chemical composition of Thymus vulgaris, Thymus zygis, and Thymus hyemalis essential oils. Food Control. 2008, 19, 681-687.

[12] Ebadollahi, A.; Ziaee, M.; Palla, F. Essential oils extracted from different species of the Lamiaceae plant family as prospective bioagents against several detrimental pests. Molecules. 2020, 25: 1556

Author Response

University of Monastir, Tunisia 
Higher Institute of Biotechnology of Monastir, Tunisia 

Taif University, Saudi Arabia

Department of Biology, College of Sciences

November 1st, 2020 

Dear Reviewer 

Please find enclosed our revised manuscript entitled “Biofilm inhibition and eradication properties of medicinal plant essential oils against Methicillin-resistant Staphylococcus aureus clinical isolates”. This article has been submitted for publication in “Pharmaceuticals”. 

Thank you for the reviewer comments pertaining to our manuscript. We are happy to submit a revised version (where corrections were labeled with a blue color in the manuscript) and here is point to point reply to the comments

 Thymus zygis ssp. gracilis or sylvestris or zygis (Boiss) ; Origanum majorana L.; Rosmarinus officinalis L.

We thank the reviewer for highlighting this point. Accordingly, the Latin names of the plants were checked and modified in the material part.

Particularly is the so low concentration of Thymol, is it referable to the Thymus zygis subspecies? can authors comment on it?

We thank the reviewer for this comment. For Thymus zygis subspecies, there are two categories. the first called thymol type or thymol chemotype which is rich in thymol, however the second type such as the oil investigated in this study contain low content in thymol.

We think that the low concentration in thymol is not referable to Thymus zygis subspecies. Further, Thymus zygis. subsp. Gracilis, Thymus zygis subsp. Sylvestris and Thymus zygis subsp. Zygis showed high content of thymol (80%, 59%, 74% of thymol respectively). In addition, Thymus albicans, Thymus mastichina subsp. donyanae, Thymus mastichina subsp. Mastichina, Thymus Funkii, Thymus Longiflorus, Thymus Lotocephalus, Thymus membranaceus and other species of thymus genus do not contain or contain less than 5% of thymol.

  1. Cristina Figueiredo, José G. Barroso and Luis G. Pedro. Volatiles from Thymbra and Thymus species of the Western Mediterranean Basin, Portugal and Macaronesia. Natural Product Communications Vol. 5 (9) 2010.

Introduction

added the following references as n.11 and 12

[11] Rota, MC.; Herrera, A.; Martinez, RM.; Sotomayor, JA.; Jordan MJ. Antimicrobial activity and chemical composition of Thymus vulgaris, Thymus zygis, and Thymus hyemalis essential oils. Food Control. 2008, 19, 681-687.

[12] Ebadollahi, A.; Ziaee, M.; Palla, F. Essential oils extracted from different species of the Lamiaceae plant family as prospective bioagents against several detrimental pests. Molecules. 2020, 25: 1556

We thank the reviewer for this comment. The proposed references were added as n. 9 and 12 since the publication of Ebadollahi et al. is about the insecticidal effect of oils.

We hope that these responses are satisfactory, and that you will now find this manuscript suitable for publication in “Pharmaceuticals”.

Looking forward to hearing from you,

Yours faithfully,

Round 2

Reviewer 1 Report

Please make the following amendments to the new conclusion (remove strikethrough, insert bold and change highlight to italic):

Line 316-317: The outcomes of this study support the medical application of O. majorana, T. zygis and R.
officinalis EOs for the prevention and/or treatment of MRSA infections and diseases as an alternative to
or combined to with antibiotics.

Line 317-318: These EOs, provided from Laboratoires OMEGA Pharma– Phytosun Arôms (France), are used orally and in high concentrations ("insert doses in parenthesis!") which shows corroborating their non-toxic effect.

Line 322: Further adequately in vitro testing or in vivo preclinical experiments trials are needed warranted to establish safety, efficacy, potential adverse effects, and interaction with other...

Author Response

University of Monastir, Tunisia 
Higher Institute of Biotechnology of Monastir, Tunisia 

Taif University, Saudi Arabia

Department of Biology, College of Sciences

November 3st, 2020 

Dear Reviewer 

Please find enclosed our revised manuscript entitled “Biofilm inhibition and eradication properties of medicinal plant essential oils against Methicillin-resistant Staphylococcus aureus clinical isolates”. This article has been submitted for publication in “Pharmaceuticals”. 

Thank you for the reviewer comments pertaining to our manuscript. We are happy to submit a revised version (where corrections were labeled with a blue color in the manuscript) and here is a point to point reply to the comments

Comments and Suggestions for Authors

Please make the following amendments to the new conclusion (remove strikethrough, insert bold and change highlight to italic):

Line 316-317: The outcomes of this study support the medical application of O. majorana, T. zygis and R. officinalis EOs for the prevention and/or treatment of MRSA infections and diseases as an alternative to or combined to with antibiotics.

Line 317-318: These EOs, provided from Laboratoires OMEGA Pharma– Phytosun Arôms (France), are used orally and in high concentrations ("insert doses in parenthesis!") which shows corroborating their non-toxic effect.

Line 322: Further adequately in vitro testing or in vivo preclinical experiments trials are needed warranted to establish safety, efficacy, potential adverse effects, and interaction with other...

 We thank the reviewer for bringing our attention to these points. Accordingly, these modifications were changed in the text.

We hope that these responses are satisfactory and that you will now find this manuscript suitable for publication in “Pharmaceuticals”.

Looking forward to hearing from you,

Yours faithfully,
